# ProteinRPN: Towards Accurate Protein Function Prediction with Graph-Based Region Proposals

## Abstract

Accurately predicting protein functions remains a significant challenge due to the intricate interplay of sequences, structures, and functions. These relationships, shaped by the principles of physics and evolutionary pressures, highlight the inherent complexity of biological systems. Recent advances in deep learning techniques demonstrate limitations in capturing the functional significance of key residues, as they predominantly rely on posthoc analyses or global structural features, resulting in suboptimal performance. Motivated by these limitations, we introduce the Protein Region Proposal Network (ProteinRPN), the first framework designed for accurate protein function prediction which seamlessly integrates functional residue identification into the prediction pipeline. ProteinRPN features a function region proposal module that identifies potential functional regions (anchors) by leveraging secondary structure definitions and spatial proximity. These anchors are refined through specialized attention mechanisms and further processed via a Graph Multiset Pooling layer. The model is trained on perturbed protein structures using supervised contrastive (SupCon) and InfoNCE losses, enabling effective modeling of residue spatial clustering and functional roles. Notably, it improves the AUPR metric by 15.4% for Biological Process (BP), 8.5% for Cellular Component (CC), and 1.3% for Molecular Function (MF) ontologies, respectively. These results highlight its efficacy in capturing the functional relevance of key residues and advancing protein function prediction.

## 1 Introduction

Advancements in genomics technology have illuminated the study of protein functions, enabling researchers to uncover the roles and interactions of proteins within living systems, making this a pivotal task in modern biology. Despite the vast number of proteins available, only a few of them have been reviewed by human curators. Among these reviewed proteins, less than 19.4% are substantiated by wet-lab experimental evidence uni (2023). Precise functional annotations of proteins are crucial for tasks such as pinpointing drug targets, unraveling disease mechanisms, and enhancing biotechnological applications across industries Kulmanov et al. (2024).

Currently, Gene Ontology (GO) Aleksander et al. (2023); gen (2021) stands out as the most comprehensive resource, embodying all the essential attributes of an ideal functional classification system. The GO consortium delineates the functional attributes of genomic products, including genes, proteins, and RNA. Specifically, GO utilizes three subontologies to organize function terms according to each product: Biological Process (BP), Molecular Function (MF), and Cellular Component (CC). Although the UniProtKB/Swiss-Prot database records manually curated GO annotations that are verified by wet-lab experiments, there are still a significant number of protein sequences lacking functional annotations due to the high costs and limited throughput of experimental studies.

To address this gap, machine learning methods have emerged as promising tools. Recently developed machine learning methods leverage different protein information for function prediction, including protein sequential information, protein tertiary structure, protein-protein interaction (PPI) networks, phylogenetic analysis, and literature information You et al. (2018; 2021; 2019); Gligorijević et al. (2021); Lai & Xu (2022); Kulmanov & Hoehndorf (2022); Kulmanov et al. (2018);

Pan et al. (2025); Kulmanov et al. (2024); Gu et al. (2023). Specifically, early studies focused on learning the similarities of homologous proteins by utilizing sequence alignment tools Gong et al. (2016). This idea was then extended to harness additional protein information, such as PPI networks and biophysical properties, to predict protein function Cho et al. (2016); You et al. (2021; 2019); Pan et al. (2025); Cho et al. (2016). However, the sequential similarity of proteins alone cannot fully determine protein function. Furthermore, these knowledge-based models heavily rely on selected features and cannot be generalized to new proteins due to the absence of prior knowledge.

Subsequent studies leverage primary sequence as the main feature for function prediction Kulmanov et al. (2018); Kulmanov & Hoehndorf (2022). While the relationship between sequence and function has been extensively investigated, translating protein structure into function remains a significant challenge. Various models, notably CNNs and graph-based deep learning approaches that incorporate both structural and functional information, have been proposed to tackle these hurdles Gligorijević et al. (2021); Lai & Xu (2022); Gu et al. (2023). However, these methods often fall short in elucidating the functional significance of key residues essential for protein functionality. Most of these approaches employ post-hoc techniques, such as Gradient-based Class Activation Maps Gu et al. (2023); Gligorijević et al. (2021), to provide visual explanations of which residues contribute most to the predicted function. Yet, this retrospective analysis lacks biological insight, as it relies solely on what the model has learned during training without accounting for prior knowledge about functional residues. Moreover, these methods often result in a selection of numerous scattered residues with low specificity, diluting the focus on the truly important regions and leading to suboptimal performance.

To address these limitations, we introduce ProteinRPN, a novel model for accurate protein function prediction. Unlike previous methods that rely on post-hoc residue analysis, ProteinRPN is the first and only model to explicitly incorporate functional residue information directly into protein function prediction, recognizing that not all residues in a sequence participate in function Jeffery (2023). Our approach incorporates biological insights into the scoring functions by modeling the spatial clustering of functional residues and their preference for specific secondary structures. This approach provides more specific and biologically meaningful insights than previous methods, ensuring that predicted functional residues are localized in well-defined structural regions. Additionally, our model introduces a graph-based Region Proposal submodule to identify functional regions within proteins, detecting key residues within $k$-hop subgraphs (anchors) and refining them through a node drop pooling layer. We also adopt specialized attention mechanisms suited to protein function prediction and model subgraph extraction as a node drop pooling task, which aligns closely with protein biology. This enables our model to identify key functional residues and subgraphs, offering more specific and biologically relevant insights than previous methods. The novelty of ProteinRPN lies in its architectural design and its ability to generate highly precise functional residue predictions, advancing the field of protein function prediction. We further enhance the representation of these functional regions using a functional attention layer and employ a Graph Multiset Transformer (GMT) to convert node-level representations into comprehensive graph-level embeddings. By integrating locally emphasized interactions while preserving the global graph structure, our model achieves a nuanced balance between fine-grained residue detection and overall protein function prediction. Additionally, we leverage contrastive learning to generate similar representations for functionally related proteins while ensuring that distinct proteins have distinct representations. Unlike adjacent tasks such as community detection, which use different techniques, our subgraph extraction is specifically modeled as a node drop pooling task, aligning closely with biological priors on functional residue clustering.

The region proposal module is initially pretrained on the PDBSite dataset Ivanisenko et al. (2000), which contains functional residue annotations sourced from the Protein Data Bank (PDB) Berman et al. (2000), a widely used database of experimentally derived structural data on proteins. We then conduct extensive experiments on standard benchmark datasets Gligorijević et al. (2021); You et al. (2021); Gu et al. (2023) to ensure a fair comparison with baseline models. The results indicate significant improvements over state-of-the-art (SOTA) models, demonstrating a ∼6% improvement in protein-centric Fmax on BP and MF ontologies, 15.4% and 8.5% improvements in AUPR on BP and CC ontologies, and a 9.2% reduction in Smin for the MF ontology. Furthermore, visualizations of predicted functional residues confirm that ProteinRPN successfully identifies essential functional structures and biologically relevant regions, highlighting its potential to enhance our understanding of protein function.

## 2 RELATED WORK

Computational methods have been proposed for protein function prediction, offering a more efficient and less resource-intensive alternative to wet-lab experimental assays. The task is framed as a multiclass multilabel classification problem, where each protein can be associated with multiple GO terms. Due to the hierarchical structure of GO terms within an ontology, predicting a given term also implies predicting all its ancestor terms, adding an additional layer of complexity. Early studies Tian et al. (2004); Gong et al. (2016) leveraged query sequence-based Multiple Sequence Alignments (MSA) to predict protein GO terms. Based on the Position-Specific Scoring Matrix (PSSM), these models could identify query sequences that are more similar to sequences in the homo-functional MSA. Consequently, the protein sequence is more likely to be annotated with the target GO term.

Machine learning models have since emerged for more accurate protein function prediction by utilizing a broader range of biological features. Some methods You et al. (2018; 2019) rely on external knowledge or even the hierarchical structure of GO terms, including GO term frequency, sequence alignment, amino acid trigram, domains and motifs, biophysical properties, and PPI networks. These approaches often employ a learning to rank (LTR) Li (2011) framework for automatic function prediction. Sequence-based methods Fa et al. (2018); Kulmanov et al. (2018); Wang et al. (2023) utilize sequential models like 1D CNNs and Transformers to derive protein sequence representations. Given that Graph Neural Networks (GNNs) are well-suited for learning the topology of PPI networks, subsequent studies Zhao et al. (2022) have combined hybrid features from protein sequences and PPI networks, embedded using GNN modules, for function prediction.

Since protein structures determine essential biological and chemical properties Jeffery (2023), relying exclusively on sequence-based methodologies may present a significant limitation. Therefore, several studies have incorporated protein structures for more accurate predictions Gligorijević et al. (2021); Lai & Xu (2022); Gu et al. (2023). Specifically, these models derive contact maps from protein structures to construct residue graphs. Additionally, as protein amino acid sequences are similar to natural language sentences, recent studies Gu et al. (2023) utilize advanced protein language models like ESM-1b Rives et al. (2021) to obtain richer sequence representations. However, there remains a gap in models that accurately detect and predict constellations of amino acids in protein active sites and leverage these for structural and functional insights Jeffery (2023).

## 3 METHODOLOGY

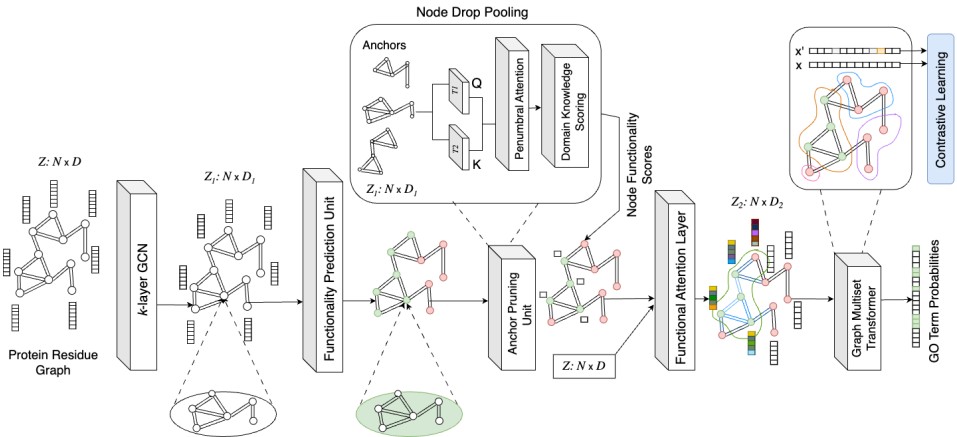

Figure 1: The ProteinRPN model predicts protein function by converting protein sequences into residue graphs using ESM Embeddings and contact maps, processing them through a k-layer GCN to identify functional subgraphs (anchors), refining these subgraphs via domain knowledge and hierarchy-aware attention mechanisms, and categorizing them into GO terms.

In this section, we introduce ProteinRPN, a novel model for protein function prediction. As illustrated in Figure 1, ProteinRPN operates on protein graphs where nodes represent individual

residues and edges are defined by the contact map which reflects residue proximity within the three-dimensional structure. The architecture is composed of three primary components. The first component, Region Proposal Network, is responsible for processing the protein graphs and proposing subgraphs which contain functionally relevant regions. These subgraphs are then fed through Functional Attention Layer which enhances the region proposals and selectively amplifies the representations of functional regions through a learned attention mechanism. The refined representations are subsequently passed to the Function Prediction block, consisting of a Graph Multiset Transformer (GMT) pooling layer and an MLP readout layer which generates predictions for GO terms. The entire framework is optimized through a combination of Supervised Contrastive (SupCon) loss and a self-supervised Information Noise-Contrastive Estimation (InfoNCE) loss, ensuring robust and effective protein representation learning.

### 3.1 MOTIVATION

Our architecture is motivated by an analysis of 603 protein structures from PDBSite Ivanisenko et al. (2000), which reveals that functional residues tend to cluster in three-dimensional space, even when they are not sequentially adjacent. Furthermore, in the studied sequences, each with hundreds of residues, the number of functional residues were 2% of the total residues. These observations, firstly, highlight the need to consider subgraphs, rather than individual nodes, in protein graphs, as protein function is influenced by the local environment and is usually carried by a cluster of residues, rather than isolated ones. It also suggests that aggressive pruning is necessary to accurately identify these few functional residues within graphs containing hundreds of nodes, necessitating a multi-stage pruning and refinement process. Finally, it is crucial that the pruning process preserves the subgraph structure, ensuring that the selected nodes form coherent clusters rather than being randomly scattered. This design is supported by our ablation results (Table 3), where removing the region proposal stage yields the largest degradation (approximately 6–14% across Fmax/AUPR and higher Smin), highlighting the necessity of explicitly localizing compact, function-bearing substructures. In contrast, relying solely on global attention disperses capacity across the full graph and dilutes signals from small but critical regions; concentrating message passing on RP-selected subgraphs provides a biologically grounded inductive bias that improves residue selection precision and downstream GO-term prediction.

### 3.2 PRELIMINARIES

Protein sequences are represented as graphs $G(V, E)$, where the vertices $V$ correspond to the protein residues, and the edges $E$ represent the proximity of residues in three-dimensional space. The adjacency matrix $A \in \mathbb{R}^{N \times N}$ for an $N$-residue protein graph is defined by calculating the contact map, where an edge is added between two nodes if the distance between their $C_\alpha$ atoms is less than 10 Å. In this work, we use $G(V, E)$ and $G(Z, A)$ interchangeably, where $V$ and $E$ denote the set of vertices and edge list, while $Z \in \mathbb{R}^{|V| \times D}$ and $A \in \mathbb{R}^{|V| \times |V|}$ represent the node features and adjacency matrices, respectively, and $D$ is the chosen dimension for residue features. The goal of ProteinRPN is to predict a probability vector $\hat{\mathbf{y}}_i^{(j)} \in \mathbb{R}^{l_j}$, where $l_j$ denotes the number of GO terms associated with subontology $j \in \{\text{BP}, \text{CC}, \text{MF}\}$. The vector $\hat{\mathbf{y}}_i^{(j)}$ represents the predicted probabilities for the $l_j$ GO terms, reflecting the likelihood of each protein being associated with multiple GO terms across all subontologies.

#### 3.2.1 RESIDUE FEATURES

Residue features for $N$ nodes in any protein residue graph are derived through a two-step process. First, each node is assigned ESM-2 Lin et al. (2023) embeddings $Z_E \in \mathbb{R}^{N \times D_E}$ to capture the intrinsic sequence-based information of the residues. In parallel, the residues are also label encoded according to their amino acid identities and transformed into embeddings $Z_R \in \mathbb{R}^{N \times D_R}$. These two feature sets are subsequently projected onto a common $D-$dimensional space and combined to form the final node embeddings $Z = Z_E + Z_R \in \mathbb{R}^{N \times D}$, effectively integrating both deep sequence information and basic residue identity.

### 3.2.2 ENHANCED DOMAIN KNOWLEDGE

To enhance the graph representation with domain-specific knowledge, we further extract the secondary structure of each residue for each protein using DSSP (Dictionary of Secondary Structure in Proteins Touw et al. (2015); Kabsch & Sander (1983), which is a database of secondary structure assignments for all protein entries in PDB Berman et al. (2000). Experimental evidence suggests that functional residues are more likely to be found in regions with defined secondary structures, such as alpha helices and beta sheets Bartlett et al. (2002). To align the residue coordinate information with the secondary structure data, we perform sequence alignments between DSSP-processed variants and residues with available PDB coordinates, addressing any discrepancies that arise between these data sources.

## 3.3 FUNCTIONAL REGION PROPOSAL NETWORK

Inspired by techniques used in object detection, we propose a strategy that selects and refines relevant regions within protein graphs to enhance function prediction. By targeting regions containing functional residues, which are often a small subset of the protein, this approach improves functional understanding. To the best of our knowledge, this is the first work to introduce a structured region selection mechanism in graphs, applied specifically to protein function prediction.

The proposed Region Proposal Network employs $k$ layers of Graph Convolutional Networks (GCNs) Kipf & Welling (2017) to process protein graphs $G(Z, A)$. In particular, let $H^{(0)} = Z$ represent the initial hidden node embedding matrix. The hidden embeddings $H$ are updated iteratively as $H^{(i+1)} = \text{ReLU}\left(\tilde{D}^{-0.5}\tilde{A}\tilde{D}^{-0.5}H^{(i)}W^{(i)}\right)$ where $\tilde{A} = A + I$ is the adjacency matrix with self-loops included, and $\tilde{D}$ is the diagonal degree matrix used for normalization. After $k$ message-passing layers, the final node embedding matrix $Z_1 = H^{(k)} \in \mathbb{R}^{N \times D_1}$ encapsulate information from their respective $k$-hop neighborhoods, effectively extending each node's receptive field to encompass its $k$-hop subgraph. Each node can then be designated as the representative of its corresponding $k$-hop subgraph, termed as an anchor. Consequently, this procedure transforms the original graph $G$ into a new graph $G'(Z_1, A)$, where each node in $G'$ corresponds to a subgraph in $G$. Empirical results indicate that setting $k = 2$ is sufficient to capture functional residues within proteins.

The second step in the region proposal module involves localizing regions that are likely to contain functional residues. This is formulated as a node classification task, where the goal is to predict whether the anchor centred around each node contains a 70% intersection with a functionally relevant region. More precisely, given the node embeddings $Z_1$ after $k$ GCN layers, the classification of each node $v_i$ in the transformed graph $G'(Z_1, A)$ is performed using a Graph Attention Network (GAT) convolution Veličković et al. (2018). The output for each node $v_i$ can be formulated as: $\hat{y}_i = \sigma\left(\sum_{j \in \mathcal{N}(i)} \alpha_{ij} W H_j^{(k)}\right)$, where, $\hat{y}_i$ is the predicted probability that the node $v_i$ in $G'$, which represents the $k$-hop subgraph $S_i$ in $G$, contains atleast 70% of the functional residues, $\alpha$ and $W$ are the attention scores and weight matrix, respectively, learnt by the GAT layer, $\mathcal{N}(i)$, represents the neighbors of node $i$ in the graph $G'(Z_1, A)$

Nodes predicted as functional are selected, and $k$-hop subgraphs (anchors) centered around these nodes are extracted. This results in a collection of anchors enriched for functional regions, ensuring high recall but with room for precision improvement. To address this, we introduce a pruning step that selectively retains the most functionally relevant subgraphs within the larger anchors. This pruning leverages a novel node-drop pooling layer that incorporates domain knowledge alongside a hierarchy-aware attention mechanism. Rather than relying on conventional dot product attention, we utilize penumbral cone attention Tseng et al. (2023) for modeling the inherent hierarchical relationships in proteins. These hierarchies span multiple levels, from the arrangement of secondary and tertiary structures to the organization of functional domains and motifs, all the way up to the interactions of subunits within protein complexes.

**Node Drop Pooling:** In order to obtain scores in our case, the feature embeddings extracted from these subgraphs are passed through GCN layers to obtain query and key representations: $q = \text{LeakyReLU}(GCN_1(G(Z_1, A)), k = \text{LeakyReLU}(GCN_2(G(Z_1, A)))$. These representations are then fed into a hierarchy-aware attention layer to decide which nodes to prune.

**Proximity Scores:** To compute proximity scores, we first measure the pairwise distances between each residue and all other residues, akin to constructing a contact map. Rather than applying a threshold to these distances, the proximity score $P_i$ for residue $i$ is computed by summing the inverse distances between residue $i$ and all other residues $j$, i.e., $P_i = \alpha_{ps} \sum_{j \neq i} \frac{1}{d_{ij}}$, where $d_{ij}$ represents the distance between residues $i$ and $j$, and $\alpha_{ps}$ is a scaling factor that determines the influence of proximity on the final node score. This method prioritizes residues that are closely clustered with a few others, resulting in higher scores compared to residues that are moderately close to many others, aligning with our insights from PDBSite Ivanisenko et al. (2000).

**Secondary Structure Scores:** Certain functional residues have been observed to preferentially reside in regions of defined secondary structure. For instance, Bartlett et al. Bartlett et al. (2002) reports that catalytic residues are frequently located in alpha helices (39%) and beta sheets (28%), with a lower prevalence in loops and unstructured regions. To reflect this, we assign higher predicted scores to residues within alpha helices and beta sheets.

The final node scores, derived from the combination of the three components, are converted into probabilities using a sigmoid function. Residues with the highest probabilities are identified as functional for subsequent processing.

## 3.4 FUNCTIONAL ATTENTION LAYER

Once candidate functional residues are identified, their representations are refined through a functional attention layer. This layer assigns weights to edges based on their connectivity to predicted functional nodes, allowing the model to emphasize relationships critical to protein function. By incorporating multistage refinement, we iteratively enhance the accuracy of functional node identification. The edge-centric approach helps preserve the structural integrity of selected subgraphs, avoiding the fragmentation that can occur when individual nodes are selected in isolation, in line with insights from PDBSite.

We feed the original residue features $Z \in \mathbb{R}^{N \times D}$ as the node feature matrix for enrichment. For each edge $(i, j)$ in the graph, the model computes an attention score $e_{ij}$ using the concatenation of the feature vectors $Z_i$ and $Z_j$, followed by a learnable weight vector $a \in \mathbb{R}^{2D_1 \times 1}$ and a ReLU activation function, that would reduce all negative scores to zero, i.e., $e_{ij} = \text{ReLU}\left(a^\top [Z_i \, \| \, Z_j]\right)$. This attention score is then adjusted based on the node type $z_j \in \{0, 1\}$ of the target node $j$, modifying the score as: $e_{ij} = \alpha_{FA} \cdot e_{ij} \cdot z_j + \beta_{FA} \cdot e_{ij} \cdot (1 - z_j)\cdot$, where $\alpha_{FA} \geq 1$ and $\beta_{FA} < 1$. This adjustment increases the attention for functional nodes while reducing it for contextual ones. For the purpose of this study, we use $\alpha_{FA} = 1, \beta_{FA} = 0.5$ in order to explicitly ensure focus on functional nodes. The attention coefficients $\alpha_{ij}$ are obtained by normalizing $e_{ij}$ across all neighbors. They determine how much influence a neighboring node $i$ has on the target node $j$.

Finally, the updated feature vector for node $j$, $Z_{2j}$, is computed by aggregating the messages from its neighbors applying, weighted by the corresponding attention coefficients, i.e., $Z_{2j} = \sum_{i \in \mathcal{N}(j)} \alpha_{ij} \cdot W \cdot Z_i$, where the transformation matrix $W \in \mathbb{R}^{D_2 \times D}$ is a learnable parameter as in the GAT layer and helps transform the initial extracted features of the nodes (residues) into a new space where relationships between residues can be more effectively captured. As a result of this operation, subgraphs surrounding functional residues—those likely to be critical for protein function—get more attention and influence the final node representations more significantly. This approach enhances the model's ability to capture the rich, context-aware interactions between residues, leading to a more comprehensive understanding of the protein's functional regions.

**Graph Multiset Transformer:** In the final step, the enriched representations $Z_2$ are fed into a Graph Multiset Transformer (GMT) layer, which transforms node-level embeddings into a comprehensive graph-level representation by capturing both local interactions and global structure. The GMT layer introduces learnable super-nodes to capture long-distance structural information and aggregates this information into a unified graph representation.

## 3.5 OPTIMIZATION FRAMEWORK

Our model is optimized using a multi-component loss function that integrates cross-entropy loss $\mathcal{L}_{\text{CE}}$ for multilabel classification, contrastive loss $\mathcal{L}_{\text{con}}$.

The contrastive loss, $\mathcal{L}_{\text{con}}$, is a combination of supervised contrastive (SupCon) loss Khosla et al. (2021) and self-supervised noise contrastive estimation (InfoNCE) loss van den Oord et al. (2019). SupCon encourages the model to cluster representations of proteins with similar GO terms, while InfoNCE ensures that the representations are robust to noise by maximizing the similarity between original and perturbed embeddings. The combined contrastive loss for a batch of $B$ proteins is defined as:

$$
\mathcal{L} = \left( -\frac{1}{B} \sum_{i=1}^{B} \sum_{j \neq i} \mathbf{1}\{y_i \cap y_j \neq \emptyset\} \cdot \log \frac{\exp(\text{sim}(z_i, z_j)/\tau)}{\sum_{k \neq i} \exp(\text{sim}(z_i, z_k)/\tau)} \right)
$$

$$
\cdot \, \alpha_{\text{SupCon}} + \left( -\log \frac{\exp(\text{sim}(z_i, z_i')/\tau)}{\sum_{z_j} \exp(\text{sim}(z_i, z_j)/\tau)} \right) \alpha_{\text{NCE}}
$$

where $z_i$ and $z_j$ are the embeddings of proteins $i$ and $j$, $\text{sim}(z_i, z_j)$ represents their cosine similarity, and $\tau$ is a temperature parameter. The indicator function $\mathbf{1}\{y_i \cap y_j \neq \emptyset\}$ ensures that only pairs with shared GO terms contribute to the SupCon loss, im order to adapt it to the multilabel case. The InfoNCE loss optimizes the similarity between the original and perturbed embeddings $z_i$ and $z_i'$. The hyperparameters $\alpha_{\text{SupCon}}$ and $\alpha_{\text{NCE}}$ control the contributions of the SupCon and InfoNCE losses.

## 4 EXPERIMENTS

### 4.1 DATASETS

PDBSite Ivanisenko et al. (2000) is a comprehensive dataset comprising biologically active sites derived from the Protein Data Bank (PDB) Berman et al. (2000). The dataset encompasses 4,723 active sites belonging to 197 different functions located within 603 proteins. PDBSite stands out among annotation databases due to its diverse representation of functional categories, enabling broad analysis across various protein functions. We leverage PDBSite to guide our model architecture and pretrain the model on predicting functional sites.

For protein function prediction, we utilize a dataset curated by Gu et al. (2023), originally developed to train their model, HEAL, which serves as our baseline. This dataset is an adapted version of the DeepFRI dataset Gligorijević et al. (2021), comprising 36,629 sequences sourced from the PDB database Berman et al. (2000) and 42,994 from the SWISS-MODEL repository Bienert et al. (2016). Further details can be found in the appendix.

### 4.2 EXPERIMENTAL SETUP

We begin by training ProteinRPN on the PDBSite which is split into training and validation sets with an 80:20 ratio, with the goal of predicting all functional sites within a protein. We extracted 603 proteins from PDBSite having 4723 functional sites with an 80-20 split of proteins for the training of stage-1. The training set for the second stage (HEAL dataset) consists of 68,078 proteins. To remove redundancy between the pretraining dataset and the downstream test set, we conducted homology filtering using MMseqs2 with stringent criteria (minimum 40% sequence identity and 80% coverage), and 2% proteins were identified as overlapping between the two datasets and removed. The details of the pretraining can be found in the appendix. Then we train the entire framework on the comprehensive protein function prediction task using the HEAL dataset.We conducted extensive experiments benchmarking ProteinRPN against state-of-the-art baselines and also evaluated leading protein language models (PLMs) augmented with a single fully connected layer for GO-term classification (Table 1). We perform ablation studies to assess the significance of each component of the model, including the impact of secondary structure, coordinate information, and contrastive learning losses, as well as test the efficacy of other modules. The results can be found Table 3 in the appendix.

The predictions of the model are evaluated using the standard Critical Assessment of Functional Annotation (CAFA) evaluator Jiang et al. (2016). Protein-centric Fmax, the maximum F1 score over

all prediction thresholds ranging from 0 to 1 with a step size of 0.1, is utilized. Smin, representing the semantic distance between the predicted and actual annotations, considers the information content of each function. The function-centric AUPR is employed as a robust measure for situations with high class imbalance. Further details on formulas and implementation are available in Jiang et al. (2016), and comprehensive information on model training and hyperparameters can be found in the appendix.

## 5 RESULTS AND ANALYSIS

### 5.1 GO TERM PREDICTION

Table 1 presents the performance metrics of ProteinRPN in comparison to all baseline models on the HEAL dataset. ProteinRPN consistently outperforms the baselines across all metrics, showing notable improvements over the HEAL model. Specifically, ProteinRPN achieves higher Fmax scores, with gains of 6.4% in Biological Process (BP), 2.7% in Cellular Component (CC), and 6.5% in Molecular Function (MF) ontologies. Beyond Fmax, ProteinRPN also demonstrates superior performance in Smin−1.78% for BP, 0.87% for CC and 9.21% for MF, and Area Under the Precision-Recall Curve (AUPR)−15.44% for BP, 8.52% for CC and 1.33% for MF, highlighting its effectiveness in predicting protein function GO terms. We conducted three independent runs of ProteinRPN and the two closest performing methods: HEAL and TAWFN, and found that the performance improvements are statistically significant ($p < 0.05$). Moreover the ablation study (details in appendix Table 3) reveals that both contrastive learning and the incorporation of domain knowledge positively contribute to the model's overall performance. During pretraining, the region proposal module exhibits strong performance, achieving an ROC of 0.95 in the anchor functionality prediction task and 0.85 in the pruning task. Although direct comparison is limited due to the absence of established baselines, the module's effectiveness is evident in downstream functional prediction tasks.

To further evaluate the contribution of each module, we analyzed the effect of removing the region proposal stage. Eliminating this component led to a substantial performance drop of approximately 6–14% across metrics, confirming its critical role in localizing functional regions. We also note that the functional attention layer cannot be removed in ablation studies, as it constitutes the first stage where functional residue signals are incorporated into the node feature representations. Without this layer, the downstream modules cannot operate as intended. In contrast, removing the domain knowledge scoring from the region proposal module results in only a small decrease in performance, since intermediate clusters of residues do not have direct ground truth labels for evaluation. Collectively, these findings highlight that while each component contributes meaningfully, the region proposal stage provides the most substantial performance gains.

Table 1: Baseline Comparison: Fmax, AUPR, and Smin of different methods on the designated test set; best performances are highlighted in bold, i.e., for Fmax and AUPR, we consider the highest, while for Smin we consider the lowest value

| Method | Fmax (↑) | | | AUPR (↑) | | | Smin (↓) | | |
|---|---|---|---|---|---|---|---|---|---|
| | BP | CC | MF | BP | CC | MF | BP | CC | MF |
| Blast Altschul et al. (1990) | 0.336 | 0.448 | 0.328 | 0.067 | 0.097 | 0.136 | 0.651 | 0.628 | 0.632 |
| FunFams Das et al. (2015) | 0.500 | 0.627 | 0.572 | 0.260 | 0.288 | 0.367 | 0.579 | 0.503 | 0.531 |
| DeepGO Kulmanov et al. (2017) | 0.493 | 0.594 | 0.577 | 0.182 | 0.263 | 0.391 | 0.577 | 0.550 | 0.472 |
| DeepFRI Gligorijević et al. (2021) | 0.540 | 0.613 | 0.625 | 0.261 | 0.274 | 0.495 | 0.543 | 0.527 | 0.437 |
| HEAL Gu et al. (2023) | 0.581 | 0.673 | 0.708 | 0.298 | 0.415 | 0.630 | 0.504 | 0.462 | 0.369 |
| ProtT5 Elnaggar et al. (2021) | 0.327 | 0.623 | 0.511 | 0.056 | 0.284 | 0.294 | 0.637 | 0.526 | 0.552 |
| SAProt Su et al. (2023) | 0.374 | 0.506 | 0.287 | 0.034 | 0.037 | 0.018 | 0.643 | 0.612 | 0.662 |
| ESM2 Lin et al. (2023) | 0.351 | 0.633 | 0.531 | 0.062 | 0.282 | 0.298 | 0.622 | 0.507 | 0.563 |
| DeepGO-SE Kulmanov et al. (2024) | 0.566 | 0.636 | 0.654 | 0.233 | 0.423 | 0.495 | 0.530 | 0.481 | 0.435 |
| TAWFN Meng & Wang (2024) | 0.548 | 0.609 | 0.711 | 0.279 | 0.346 | 0.674 | 0.561 | 0.539 | 0.393 |
| PFresGO Pan et al. (2025) | 0.568 | 0.674 | 0.692 | 0.293 | 0.361 | 0.602 | 0.535 | 0.498 | 0.417 |
| **ProteinRPN** | **0.618** | **0.691** | **0.754** | **0.344** | **0.459** | **0.683** | **0.495** | **0.458** | **0.335** |

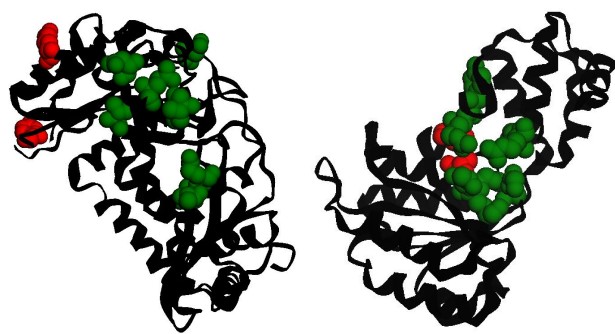

Figure 2: Visual Demonstration of Region Proposal Network detected residues in proteins (a) 2BCC-B and (b) 2CHG-A

## 5.2 FUNCTIONAL RESIDUE VISUALIZATION

We conducted extensive experiments on 120 proteins from the held-out test set of the PDBSite dataset. These proteins averaged 445 residues each, with 7.7 functional residues on average. Our model accurately detected functional residues with an 84% accuracy, compared to 75% with current methods studied on fewer proteins Jang et al. (2024). Additionally, we achieved an AUCROC of 0.8821 for functionality probability predictions across all residues, demonstrating strong generalizability. Here we present case studies by evaluating the functional residue predictor in ProteinRPN by analyzing specific proteins. For example, on protein 2BCC (B chain, 422 residues, 10 functional), ProteinRPN accurately identifies 8 functional residues, with region proposals covering subgraphs of 28 residues, as shown in Fig. 2(a). Functional residues predicted correctly are highlighted in green, while missed ones are marked in red. Similarly, for protein 2CHG (A chain, 226 residues, 11 functional), the model successfully identifies 9 functional residues, with region proposals covering 43 residues. As shown in Fig. 2(b), the correctly identified residues are closely clustered within the structure, while the missed residues are located farther from the cluster. These results demonstrate its ability to accurately identify and localize constellations of functional residues.

## 6 CONCLUSION

In this work, we introduced ProteinRPN, a novel graph-based model equipped with graph region proposal networks which is designed to identify and refine functional regions within protein residue graphs. By leveraging hierarchical attention mechanisms, domain-specific knowledge, and multistage refinement, through a combination of supervised contrastive learning and self-supervised InfoNCE loss, ProteinRPN significantly improves the accuracy of protein function prediction across GO terms. Our results demonstrate substantial gains over SOTA methods, with enhanced precision in identifying functional residues and preserving structural integrity in predicted subgraphs. While our model provides generalized insights across a range of protein functions, the present analysis is limited to a subset of protein structures. Future work will focus on extending the model's capabilities by incorporating diverse knowledge sources and exploring additional mechanisms to further enhance the accuracy and scalability of protein function prediction. We envision ProteinRPN as a foundation for integrating diverse structural and evolutionary insights, enabling more scalable and biologically faithful protein function prediction.

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

# A APPENDIX

## A.1 PROTEIN FUNCTION PREDICTION DATASET

For protein function prediction, we utilize a dataset curated by Gu et al. Gu et al. (2023), originally developed for training their HEAL model, which serves as our baseline. This dataset is a modified version of the DeepFRI dataset Gligorijević et al. (2021), containing sequences sourced from the PDB database and the SWISS-MODEL repository. Each dataset entry includes contact maps, where residues are considered in contact if the distance between their $C_\alpha$ atoms is less than 10 Å, and ESM-1b embeddings for individual residues. The dataset is split into training, validation, and test sets, maintaining an 8:1:1 ratio as can be seen in Table 2.

For predicting Gene Ontology (GO) terms in alignment with standard practices, we use the leaf nodes of the directed acyclic graph as prediction labels, resulting in 489 binary prediction tasks for Molecular Function (GO-MF), 1,943 for Biological Process (GO-BP), and 320 for Cellular Component (GO-CC).

We employ a multi-cutoff split method, as utilized by Gligorijevic et al. (2021), ensuring that the test set comprises only PDB chains with sequence identities of 95% or less. Furthermore, each PDB chain in the test set is guaranteed to have at least one experimentally validated GO term for each GO domain. The test set remains consistent with that used by DeepFRI and other baseline models to maintain parity of comparison.

| Dataset | Train | Val | Test |
|---|---|---|---|
| PDB | 29,893 | 3,322 | 3,414 |
| SWISS-MODEL | 38,185 | 4,242 | 567 |

Table 2: Dataset distribution for training, validation, and testing.

In order to train the ProteinRPN model, we combine the training and validation sequences from the PDB and SWISS-MODEL repositories. To maintain consistency with the DeepFRI benchmark dataset (Gligorijevic et al., 2021), we use only the PDB test set as our test set, while the SWISS-MODEL test set is incorporated into the training set.

## A.2 REGION PROPOSAL MODULE PRETRAINING

In alignment with the region proposal training procedure for downstream tasks, we treat each node's $k$-hop subgraph as an anchor, where the node itself serves as the representative of the anchor. This reformulates the original graph $G$ into a transformed graph $G'(H', A)$, where each node in $G'$ corresponds to a subgraph in $G$. Two Graph Neural Network (GNN) layers are applied to $G'$. The first layer predicts the likelihood of each anchor containing a functional site, while the second generates a vector determining which nodes within the anchor's subgraph should be retained or pruned.

Anchors are labeled as positive based on their Jaccard Similarity overlap with ground-truth annotations. Specifically, an anchor is considered positive if its Jaccard Similarity exceeds a threshold of 0.7, and negative if it falls below 0.3. This dual-threshold strategy ensures a clear separation between functionally relevant and irrelevant regions.

The optimization of ProteinRPN is guided by a loss function that combines classification and regression tasks, inspired by Faster R-CNN:

$$\mathcal{L}(\{p_i\}, \{t_i\}) = \frac{1}{N_{\text{cls}}} \sum_i \mathcal{L}_{\text{cls}}(p_i, p_i^*) + \lambda \frac{1}{N_{\text{reg}}} \sum_i p_i^* \mathcal{L}_{\text{pruning}}(G_i, G_i^*)$$

where:

- $\mathcal{L}_{\text{cls}}(p_i, p_i^*)$ is the classification loss, computed as binary cross-entropy between the predicted probability $p_i$ and the ground truth $p_i^*$, indicating the presence of a functional site.
- $\mathcal{L}_{\text{pruning}}(G_i, G_i^*)$ measures pruning accuracy, defined by the discrepancy in overlap between pruned and ground-truth subgraphs.

- $\lambda$ is a balancing factor regulating the relative contributions of classification and pruning losses. Through experimentation, $\lambda = 1$ was found to offer an optimal tradeoff between identifying functional sites and accurately delineating their spatial boundaries.

This structured loss function, inspired by that of Faster R-CNN Ren et al. (2015), facilitates precise refinement of functional site predictions and their boundaries within the protein structure. High-confidence anchors and their associated subgraphs are aggregated, with overlapping subgraphs being unified to produce comprehensive functional annotations. Specifically, anchors with a high probability of containing functional regions (probability of functionality $> 0.7$) are identified. Within these anchors, residues with high retention probabilities (probability of retention $> 0.5$) are retained as functional residues. Finally, functional residues from nearby high-confidence anchors are unified.

### A.3 ABLATION STUDIES

As detailed in Section 5.1, the ablation results demonstrate that both contrastive learning and domain knowledge yield consistent gains in performance. The region-proposal module is the primary driver of performance: during pretraining it attains ROC scores of 0.95 (anchor functionality) and 0.85 (pruning), and its removal induces a 6–14% decrease across evaluation metrics, confirming its central role in localizing functional regions. By contrast, disabling domain-knowledge scoring within the proposal stage produces only a minor degradation, since intermediate clusters of residues do not have direct ground truth labels for evaluation. Together these findings indicate that while all components contribute, the region-proposal stage delivers the primary benefit, as can be seen in Table 3.

Table 3: Ablation Studies: Fmax, AUPR, and Smin of different variants of ProteinRPN, where CL: Contrastive Learning, SS: secondary structure and proximity scoring, RP: Region Proposal stage. Best performances are highlighted in bold, i.e., for Fmax and AUPR, we consider the highest, while for Smin we consider the lowest value.

| Model | Fmax ($\uparrow$) | | | AUPR ($\uparrow$) | | | Smin ($\downarrow$) | | |
|---|---|---|---|---|---|---|---|---|---|
| | BP | CC | MF | BP | CC | MF | BP | CC | MF |
| ProteinRPN CL | **0.6175** | **0.6906** | **0.7542** | **0.3438** | 0.4527 | **0.6833** | **0.4948** | **0.4576** | **0.3350** |
| ProteinRPN w/o CL | 0.6009 | 0.6878 | 0.7408 | 0.3223 | 0.4166 | 0.6479 | 0.5062 | 0.4587 | 0.3557 |
| ProteinRPN w/o SS w CL | 0.6114 | 0.6894 | 0.7498 | 0.3426 | **0.4591** | 0.6778 | 0.4984 | **0.4576** | 0.3421 |
| ProteinRPN w/o SS w/o CL | 0.5975 | 0.6801 | 0.7364 | 0.3161 | 0.4242 | 0.6446 | 0.5088 | 0.4674 | 0.3547 |
| ProteinRPN w/o RP | 0.5810 | 0.6730 | 0.7080 | 0.2980 | 0.4150 | 0.6300 | 0.5040 | 0.4620 | 0.3690 |

To supplement our ablation studies and assess the importance of modules that could not directly be removed, we investigate the efficacy of these individual components by replacing them with alternatives. Specifically, we explore a defined combination of our existing contrastive learning objectives, SINCERE Feeney & Hughes (2024), and substitute the Graph Multiset Transformer pooling layer module with other variants of graph transformers such as Polynormer Deng et al. (2024) to evaluate their impact on model performance.

### A.3.1 SINCERE LOSS

For our contrastive learning objective, we employed a combination of supervised contrastive loss (SupCon) and InfoNCE. Recent literature suggests advanced formulations that integrate both losses in a unified framework. One such formulation is the Supervised Information Noise-Contrastive Estimation (SINCERE) loss Feeney & Hughes (2024). SINCERE is an extension of InfoNCE, specifically adapted for supervised learning, and addresses the issue of within-class repulsion seen in SupCon by maximizing within-class similarity.

For each protein graph $G$, we generate multiple views by perturbing node embeddings with random noise. SINCERE aims to maximize the similarity between views of the same protein while minimizing similarity across different proteins. Mathematically, the loss is defined as:

$$\mathcal{L}_{\text{SINCERE}} = -\frac{1}{M} \sum_{m=1}^{M} \log \frac{\exp(\text{sim}(\mathbf{z}_m, \mathbf{z}'_m)/\tau)}{\sum_{m'=1}^{M} \exp(\text{sim}(\mathbf{z}_m, \mathbf{z}_{m'})/\tau)}$$

where $\text{sim}(\mathbf{z}_m, \mathbf{z}'_m)$ represents the cosine similarity between two augmented views of the $m$-th protein graph, and $\tau$ is the temperature parameter set to 0.1. This contrastive framework enhances the model's ability to learn robust and discriminative graph representations by promoting similarity for functionally related regions while ensuring distinct representations for different proteins.

We replace our usual contrastive loss with SINCERE and combine it with binary cross-entropy (BCE) loss for classification. However, our experiments reveal that the combination of SupCon and InfoNCE outperforms using either InfoNCE alone or SINCERE as can be seen in Table 4. This is likely because the external combination of SupCon and InfoNCE offers more fine-grained control over their relative importance, while SINCERE inherently predefines their integration.

### A.3.2 POLYNORMER: A POLYNOMIAL-EXPRESSIVE GRAPH TRANSFORMER

In a final experiment, we assess the efficacy of the Graph Multiset Transformer (GMT) module by substituting it with an empirically effective graph transformer, Polynormer Deng et al. (2024).

Polynormer Deng et al. (2024) is designed to balance expressivity and scalability in graph learning tasks. Traditional GNNs often suffer from over-smoothing and limited expressive power when modeling complex functions. Polynormer addresses these issues by learning high-degree polynomials on graph data, enabling it to capture intricate node relationships while maintaining linear computational complexity. Formally, node representations at layer $l$ are computed as:

$$X^{(l)} = \left(W^{(l)} X^{(l-1)}\right) \odot \left(X^{(l-1)} + B^{(l)}\right)$$

where $W^{(l)} \in \mathbb{R}^{n \times n}$ and $B^{(l)} \in \mathbb{R}^{n \times d}$ are trainable weight matrices, $\odot$ denotes the Hadamard product, and $X^{(l-1)}$ is the input node feature matrix at layer $l-1$. Polynormer models polynomial functions where the degree of the polynomial grows exponentially with the number of layers, allowing a depth $L$ Polynormer to represent polynomials of degree $2^L$.

Polynormer integrates graph structure through two types of equivariant attention mechanisms: local attention, which incorporates adjacency information, and global attention, which captures higher-order interactions across the entire graph. This local-to-global attention mechanism mirrors the intuition behind using GMT, making Polynormer a suitable candidate for comparison.

However, in Table 4 we observe a significant performance drop when substituting GMT with Polynormer. This highlights the effectiveness of GMT's supernode-based topological pooling over traditional pooling approaches that treat nodes equally. The introduction of supernode representations in GMT proves to be more adept at capturing key functional substructures, which are critical for accurate function prediction.

| Model | Fmax (↑) | | | AUPR (↑) | | | Smin (↓) | | |
|---|---|---|---|---|---|---|---|---|---|
| | BP | CC | MF | BP | CC | MF | BP | CC | MF |
| ProteinRPN CL | **0.6175** | **0.6906** | **0.7542** | **0.3438** | **0.4527** | **0.6833** | **0.4948** | **0.4576** | **0.3350** |
| ProteinRPN w SINCERE | 0.5823 | 0.6676 | 0.7513 | 0.3128 | 0.3990 | **0.6833** | 0.5180 | 0.4779 | 0.3461 |
| ProteinRPN w Polynormer | 0.5102 | 0.6269 | 0.5877 | 0.2012 | 0.3076 | 0.4030 | 0.5629 | 0.5257 | 0.4908 |

Table 4: Fmax, AUPR, and Smin of different variants of the proposed model. Best performance in bold where applicable.

### A.4 TRAINING SETUP

We train the proposed ProteinRPN model using the Adam optimizer with a learning rate of 0.0001 and a batch size of 48 for 100 epochs. All models are implemented using PyTorch and the PyTorch Geometric library Paszke et al. (2017); Fey & Lenssen (2019). Training is conducted on a single NVIDIA A100 80 GB Tensor Core GPU, with training times of approximately 10 hours per model using a batch size of 48.

**Hyperparameter Settings:** All hyperparameters are tuned using Optuna Akiba et al. (2019), which employs Tree-structured Parzen Estimator (TPE) sampling. The input feature dimension is set to

$D = 1280$, and the hidden channels in the $k$-layer GCN are $D_1 = 256$ with $k = 2$. The output dimension for the functional attention layer is $D_2 = 512$. The loss function tuning parameters are set to $\alpha_{\mathrm{cc}} = 0.001$, $\alpha_{\mathrm{SupCon}} = 0.01$, and $\alpha_{\mathrm{NCE}} = 0.01$.

