# OpenReview forum: "ProteinRPN: Towards Accurate Protein Function Prediction with Graph-Based Region Proposals"
_ICLR.cc/2026/Conference — Submitted to ICLR 2026_

### Official Review · Reviewer_J7zo · 2025-10-26

**Soundness:** 2
**Presentation:** 3
**Contribution:** 2
**Rating:** 2
**Confidence:** 3

**Summary:**

To improve the accuracy of identifying functional residues, the authors propose a model called ProteinRPN. This architecture integrates hierarchical attention mechanisms, domain-specific knowledge, and multistage refinement. Compared to state-of-the-art baselines on the PDBSite datasets, ProteinRPN demonstrates superior performance.

**Strengths:**

The prediction of protein functions is a valuable research direction in bioinformatics. The proposed method shows promising performance on the PDBSite dataset.

**Weaknesses:**

- The design of the proposed method lacks elegance. The model primarily consists of several existing neural network (NN) components (e.g., GNNs and attention modules) that are combined to learn representations in a somewhat unrefined manner.
- I have significant concerns regarding the method's efficiency. The extensive use of large-scale NN components probabily lead to excessive time consumption. Achieving superior performance at the expense of lengthy processing times is not ideal. The authors should consider this and report the training time for both the proposed method and representative baselines.
- Since the designed architecture is motivated by observations from the PDBSite dataset, a more fiar performance verification should be conducted using another dataset.

**Questions:**

- I recommend that the authors clarify the contributions of their work in the introduction section to make them more concise and explicit.
- In Figure 1, what do the red and green nodes represent? Additionally, what do the colored circles surrounding a group of neighbors signify? Do they indicate the neighboring nodes of an anchor?
- In line 215, it should be clarified that \( D = D_E + D_R \).
- I do not understand the sentence, “To align the residue coordinate ... between these data sources” in line 223.
- The description of the subgraph is unclear. The statement “G’(Z_1, A), where each node in G’ corresponds to a subgraph in G” needs refinement. If the authors wish to use the concept of a “subgraph,” please provide a precise definition, such as the k-hop subgraph of \( V_i \). If each element in \( Z_1 \) is considered a representation of a subgraph, please avoid using the term "node embeddings" \( Z_1 \) again in line 250.
- Why is \( \hat{y}_i \) interpreted as functional? It seems to be merely a high-level representation of each node without any specific physical meaning. How do the authors justify the claim that \( \hat{y}_i \) has a high recall? What measures are in place to ensure this in the model?
- I do not understand how the node-drop pooling layer incorporates domain knowledge.
- I am unclear why \( p_i \) can prioritize residues that are closely clustered with only a few nodes.
- Do the "three components" refer to the proximity score and two secondary structure scores?
- Why does setting \( \alpha_{FA} = 1 \) and \( \beta_{FA} \) result in \( e_{ij} \) focusing on functional nodes?
- I suggest changing the notation from \( Z_{2j} \) to \( [Z_2]_j \).
- Please add a citation for the GMT layer.
- How many loss terms does the model utilize? Is it one cross-entropy (CE) loss and two contrastive (CON) losses? If so, please include \( L_{CE} \) in the expression for \( L \).
- The method for generating perturbation \( z’_i \) is not provided.
- A sensitivity analysis for \( \alpha_{ps} \), \( \alpha_{FA} \), \( \beta_{FA} \), and \( \tau \) is missing.

---

### Official Review · Reviewer_6b5X · 2025-10-30

**Soundness:** 3
**Presentation:** 4
**Contribution:** 3
**Rating:** 4
**Confidence:** 4

**Summary:**

Existing deep learning methods often rely on post-hoc analysis or global structural features, failing to effectively capture the functional significance of key residues.
To address this, the authors propose ProteinRPN, a novel framework that integrates functional residue identification directly into the prediction pipeline. Key components include: A Function Region Proposal Module that leverages secondary structure definitions and spatial proximity to identify potential functional subgraphs (anchors) ; A Functional Attention Layer to amplify representations of functional regions while preserving subgraph structural integrity.

**Strengths:**

1.  ProteinRPN is compared to 11 SOTA methods (e.g., HEAL, DeepFRI, ProtT5) across three GO subontologies using standard metrics (Fmax, AUPR, Smin), with statistically significant improvements confirmed via three independent runs.
2. By accurately localizing functional subgraphs, it helps uncover the link between protein structure and function—critical for understanding enzyme catalysis, ligand binding, and disease-related mutations.
3. Case studies (e.g., proteins 2BCC and 2CHG) visually confirm that predicted functional residues cluster in 3D space, reinforcing the model’s biological relevance. It is good to see such cases.

**Weaknesses:**

1. The model is evaluated exclusively on the HEAL dataset (a modified version of DeepFRI). To confirm generalizability, it should be tested on independent datasets with different characteristics, such as:
CAFA Challenge Datasets: Standard benchmarks for protein function prediction, which include proteins with sparse annotations (mimicking real-world scenarios).
Human-Specific Datasets: Proteins with well-characterized functions (e.g., from UniProtKB/Swiss-Prot) to validate performance on clinically relevant targets.

2. There is only one task in this paper, only including GO term prediction. I am afraid that this model can only predict GO term, although we can use different functional labels to train this model again.

3. Enhance Interpretability with Attention Visualization: The Functional Attention Layer assigns weights to edges, but the paper does not visualize these weights. Add heatmaps showing attention scores between functional and non-functional residues to confirm that the model prioritizes biologically meaningful interactions (e.g., residues in active sites).

**Questions:**

Gene Ontology is hierarchical (predicting a term implies predicting its ancestors), but the paper does not clarify if ProteinRPN explicitly enforces this hierarchy. Does the model account for parent-child relationships between GO terms, or does it treat each term as independent (risking inconsistent predictions)?

---

### Official Review · Reviewer_YrZ7 · 2025-10-31

**Soundness:** 2
**Presentation:** 3
**Contribution:** 2
**Rating:** 4
**Confidence:** 2

**Summary:**

This paper introduces ProteinRPN, a novel graph-based model for protein function prediction. The core innovation is the integration of a Region Proposal Network (RPN), inspired by object detection, which aims to explicitly identify functionally relevant subgraphs (regions) within the protein structure, rather than relying on post-hoc analysis of a model trained on global features. This module is pre-trained on the PDBSite dataset to identify known functional sites. The architecture then uses this module to propose and prune "anchors" (k-hop subgraphs) based on learned features and injected domain knowledge (secondary structure and residue proximity). These refined regions are processed by a functional attention layer and a Graph Multiset Transformer (GMT) to produce final GO term predictions. The model is trained with a combination of cross-entropy and contrastive losses (SupCon and InfoNCE). The authors report state-of-the-art results on the HEAL dataset benchmark, outperforming previous models like HEAL and DeepFRI.

**Strengths:**

1. The central motivation is strong and well-argued. Moving beyond post-hoc explanations of residue importance (like CAMs) and integrating functional site identification directly into the prediction pipeline is a novel and sensible contribution to the field.

2. The empirical results presented in Table 1 are compelling, showing consistent improvements over several strong baselines (including HEAL and DeepFRI) across all three GO ontologies and multiple metrics (Fmax, AUPR, Smin).

3. The use of the PDBSite dataset to pre-train the region proposal module is a logical and well-justified methodological choice, grounding the model's region-finding capability in experimental data before the main function prediction task.

4. The ablation studies (Table 3) effectively demonstrate the contribution of the key components, particularly the significant performance drop when the Region Proposal (RP) stage is removed, which validates its critical role in the model's success.

**Weaknesses:**

1. The most significant weakness is the potentially unfair experimental comparison. ProteinRPN is pre-trained on PDBSite (functional residues), but the baselines (e.g., HEAL) are not. The performance gain may stem from this extra supervision, not the RPN architecture. A fair comparison would require pre-training the baselines similarly.

2. The model architecture is overly complex, combining numerous recent techniques (k-hop GCNs, GAT, penumbral cone attention, GMT, SupCon, InfoNCE). This "kitchen sink" approach makes it difficult to disentangle.

3. The description of the Region Proposal module's mechanics is convoluted. It seems to use multiple GNNs and attention steps (a GCN for anchors, a GAT for classification, then another GCN/attention for pruning). The justification for this complexity over a simpler scoring mechanism is unclear.

4. The "domain knowledge" scoring (proximity, secondary structure) feels ad-hoc. Assigning fixed scores based on structure is a potentially biasing heuristic, as many functional residues exist outside these regions. The ablation table's (Table 3) naming for this is also confusing

**Questions:**

1. During RPN pre-training, anchors are labeled positive (> 0.7 Jaccard) or negative (< 0.3). What happens to anchors in the 0.3-0.7 "limbo" region? Are they ignored?

2. Was homology filtering performed between PDBSite and the HEAL training/validation sets, not just the test set? Any overlap here would constitute data leakage.

3. In Figure 1, what is the role of the "Functionality Prediction Unit" block, and how does it differ from the "Node Drop Pooling" that generates "Node Functionality Scores"? The sequence is unclear.

---

### Official Review · Reviewer_cLWm · 2025-11-01

**Soundness:** 3
**Presentation:** 3
**Contribution:** 2
**Rating:** 4
**Confidence:** 4

**Summary:**

The authors introduces ProteinRPN, a graph-based model for protein function prediction that explicitly integrates functional residue identification into the prediction pipeline the integrates Region Proposal Networks (RPNs) adapted for graphs to identify functional subgraphs (“anchors”) based on 3D proximity and secondary structure, a node-drop pooling mechanism guided by hierarchy-aware attention, a functional attention layer that emphasizes residues likely to participate in biochemical activity, a Graph Multiset Transformer (GMT) for graph-level readout, and a multi-objective training setup combining Supervised Contrastive (SupCon) and InfoNCE losses.

Empirically, the model shows notable improvements (up to +15.4% AUPR) over SOTA baselines (DeepFRI, HEAL, PFresGO, etc.) on GO term prediction tasks, and achieves 84% accuracy in identifying functional residues.

**Strengths:**

The main strength of the paper is a strong motivation of the underlying biological problem and the integration of domain knowledge.

1. The inclusion of functional residue localization as the learning objective highlights the biological focused application.
2. The adoption of RPN concept is novel in this context.
3. strong empirical performance gain compared to other methods on this problem.

**Weaknesses:**

The main weakness of the paper is clarity and limited justification for many of the specific design choices.

1. The paper will benefit from a more systemic construction of the methods motivated by each of the strength since this method is essentially a composition of existing architectural components. Formalizing the RPN adaptation mathematically and show the inductive bias introduced by teh anchor subgraph proposal etc.

2. There is not theoretical justification for the regional proposals, results comparing with other subgraph sampling method such as Top-k attention or random region will be helpful.

3. The new contrastive objective can use some more analysis and justification. The choice for multiple coefficients are not justified either. The current ablation of the whole contrastive objective shows small improvement, the dual contrastive objective needs more justification as well.

**Questions:**

1. What makes the proposed module fundamentally different from existing subgraph selection or graph pooling methods (e.g., SAGPool, ASAP, or DiffPool)?

2.You define an anchor as a node’s k-hop subgraph and predict if it overlaps ≥ 70% with a functional region. Why 70%? Was this threshold tuned empirically or derived from biological reasoning?

3. in the hierarchy-aware penumbral cone attention pruning,  what is learned here versus what is fixed by hierarchy priors? Does the cone attention directly model GO term hierarchy or only local structural hierarchy?

4. For the dual contrastive objectives, Why are both needed? Did you test SupCon-only and InfoNCE-only baselines quantitatively?

---

### Meta-Review · Area_Chair_X17v · 2026-01-07

**Summary:**

The paper proposes a new approach to protein function prediction. The method combines deep learning with biologically inspired heuristics. There are several highlighted components of the model: region proposal network, specialized attention mechanism, attention pooling, pre-training with contrastive-style losses. The model is evaluated on the HEAL benchmark and shows state-of-the-art performance.

Here are the key pros and cons based on the reviews and the paper itself:

Pros:
1. Addressing an important problem.
2. A well-engineered system that performs well on the HEAL benchmark

Cons:
1. The paper is engineering-heavy: the method mixes many components and it is not very clear how much which of them matter, even given there is some ablation study in the paper.
2. The evaluation is only on the HEAL benchmark. One of the reviewers proposes many other benchmarks and tasks.
3. The method trains on PDBSite while the baselines don't. This is in principle okay, but should be very clearly highlighted and evaluated

There was no rebuttal.

Overall, the paper definitely has merit and is promising, but has room for improvement on the thoroughness of benchmarking and analysis of the method. Therefore, at this point I recomment rejection, but encourage the authors to improve the paper and resubmit to a different venue, and/or to submit to a specialized (workshop) venue.

**Reviewer Concerns:**

Listed above. No rebuttal.

**Reviewer Scores:**

No rebuttal

---

### Decision · Program_Chairs · 2026-01-26

Reject